# Combining Abilities, Heterosis, Growth Performance, and Carcass Characteristics in a Diallel Cross from Black-Bone Chickens and Thai Native Chickens

**DOI:** 10.3390/ani12131602

**Published:** 2022-06-21

**Authors:** Piriyaporn Sungkhapreecha, Vibuntita Chankitisakul, Monchai Duangjinda, Wuttigrai Boonkum

**Affiliations:** 1Department of Animal Science, Faculty of Agriculture, Khon Kaen University, Khon Kaen 40002, Thailand; pat_sungkhapreecha@hotmail.com (P.S.); vibuch@kku.ac.th (V.C.); monchai@kku.ac.th (M.D.); 2Network Center for Animal Breeding and Omics Research, Faculty of Agriculture, Khon Kaen University, Khon Kaen 40002, Thailand

**Keywords:** body weight, average daily gain, Thai indigenous chicken, genetic

## Abstract

**Simple Summary:**

Black-bone chicken is classified as a healthy food due to its naturally bioactive ingredients; however, the limitation of a slow growth rate results in longer times to raise it to market weight. Crossbreeding is a tool used to produce superior breeds. The present study therefore determined the combining abilities and heterosis for better growth performance and maintained the color of black-bone chickens to gain desirable antioxidant properties (melanin and carnosine) in crosses between Hmong black-bone (HB), Chinese black-bone (CB), and Thai native (TN) chickens. The results indicate that crossing between the TN sires and CB dams had the best potential to promote careers for farmers on a commercial scale.

**Abstract:**

The purpose of this study was to determine the combining abilities and heterosis for the growth performance and carcass characteristics in crosses between Hmong black-bone (HB), Chinese black-bone (CB), and Thai native (TN) chickens using a mating system diallel crossing. Nine crossbred chickens including HB × HB, CB × CB, TN × TN, HB × TN, TN × HB, CB × HB, HB × CB, TN × CB, and CB × TN, were tested. The total data were 699 recorded at the beginning of the experiment to 595 recorded in weeks 14 of age. Body weight (BW), average daily gain (ADG), feed conversion ratio (FCR), and survival rate (SUR) were recorded. Heterosis and combining ability regarding general combining ability (GCA), specific combining ability (SCA), and reciprocal combining ability (RCA) were estimated. The study found that CB had the greatest BW and ADG at all weeks (*p* < 0.05) except for hatch, while those of HB were the lowest. The highest GCA was found in CB; meanwhile, GCA was significantly negative in HB of all ages. Crossing between TN × CB had the greatest BW from 8 weeks of age, which was related to positive SCA and RCA values. However, the RCA value of TN × CB was lower than the SCA value of CB × TN. The yield percentages of the carcass in CB (87.00%) were higher than those in TN (85.05%) and HB (82.91%) (*p* < 0.05). The highest breast and thigh meat lightness (*L**) values were obtained in TN (*p* < 0.05), while those of CB and HB were not different (*p* > 0.05). In the crossbreed, the yield percentage of the carcass was highest in TN × CB (89.65%) and CB × TN (88.55%) (*p* > 0.05) and was lowest in TN × HB (71.91%) (*p* < 0.05). The meat and skin color of the breast and thigh parts in the crossbreed had the lowest lightness in HB × CB (27.91 to 38.23) (*p* < 0.05), while those of TN × CB and CB × TN were insignificant (*p* > 0.05). In conclusion, crossing between the TN sires and CB dams has the preferable potential to develop crossbred Thai native chickens for commercial use based on their high growth performance.

## 1. Introduction

Improving livestock productivity is a crucial issue that has been debated as concerns about the global population growth require adequate feed production, particularly meat protein sources [1,2,3]. Population growth is the main reason for the increase in consumption, and the projected 11% global increase will support a projected 14% increase in global meat consumption by 2030 [4]. In the meat sector, poultry meat will continue to be the main growth driver in meat production. Overall, most of the growth in poultry meat production will occur in developing regions, accounting for 84% of the additional production [4,5]. Consumers prefer poultry as a low-cost source of protein [3,6]. In addition, people of all religious capital can eat poultry meat without restrictions. Finally, poultry meat is still a healthier food than pork or beef meats [7,8,9].

Global poultry meat consumption is expected to increase to 152 Mt during the forecast period, accounting for 52% of the additional meat consumption. Rapidly growing per capita consumption rates of poultry reflect its important role [10]. The average poultry meat consumption is 0.45% higher than in the previous year and 12.5% more than 10 years ago. In addition, the problems of poverty, hunger, and quality of life of the world’s population still need to be addressed. Populations in underdeveloped and developing countries are among the most likely to suffer from food shortages due to higher population growth rates than those in developed countries [11,12]. However, a way to reduce the severity of food shortages and still obtain good quality food for consumption is to improve the indigenous resources available in the area.

Native chickens are indigenous animals distributed throughout the world, especially in rural communities [13,14,15]. Many academic data support Thai native and black-bone chicken meat quality as healthy foods [16,17,18,19,20,21]. Black-bone chickens typically have black skin, meat, and bones. They have desirable bioactive compounds including melanin and carnosine, which are antioxidant properties, and both also decrease the cholesterol levels in the blood [18,19,20,21]. Variations in phenotypes through qualitative and quantitative traits can be found in Thailand [22]. However, the poor growth performance of these chicken lines is a limitation to promoting careers for farmers on a commercial scale. A few studies on genetic improvement have been conducted for this species in Thailand.

Meanwhile, Thai native chicken has been appreciated by consumers for the flavor and texture of tender meat [23,24]. In addition, the Thai native chicken line called Pradu Hang dum was developed, and their genetics were improved until satisfying the growth performance, egg production, and fertility [25,26,27,28,29], resulting in promotion by the Network Center for Animal Breeding and Omics Research (NCAB), Faculty of Agriculture, Khon Kaen University, Thailand. Therefore, crossbreeding programs are preferred to utilize the sound characteristics of those chickens to possibly exploit the phenomenon of heterosis to evolve a more resistant hybrid chicken including the upgrading local types with suitable exotic ones. Crossing indigenous chicken genotypes has led to improvements in faster growth rates and higher performance such as body weight, average daily gain (ADG), feed conversion ratio (FCR), and survival rate (adaptation to harsh environmental conditions such as heat stress conditions). Combining ability and heterosis analysis is one popular method to identify acceptable parents who can combine well and produce the desired crossbred animals [30]. Combining ability and heterosis will also help us to understand the transmission of the desired genes from selected parents to their offspring, which necessitates a thorough understanding of gene function [31,32]. Furthermore, this provides us with insight into the design of appropriate breeding methods such as a diallel cross mating system and provide information regarding a parent’s general and specialized combining abilities as well as their cross combinations [33].

Therefore, the objective of this study was to evaluate the appropriate mating pairs for growth performance traits between black-boned chickens (Chinese black and Hmong) and Thai native chickens (Pradu Hang dum) to develop and provide new F1 commercial chickens for the Thai market.

## 2. Materials and Methods

### 2.1. Data Collection

The individual growth performance of Hmong black-bone (HB), Chinese black-bone (CB) (as a representative of black-bone chickens raised in Thailand), and Thai native (TN) chickens was recorded from the experimental farm of the Network Center for Animal Breeding and Omics Research, Faculty of Agriculture, Khon Kaen University, Thailand. A total of 699 chickens were used to record the body weight at birth (BW0), body weight at 4, 8, 12, and 14 (slaughtering weight) weeks of age (BW4, BW8, BW12, and BW14), average daily gain (ADG) during 0–4, 0–8, 0–12, 0–14 weeks of age (ADG0–4, ADG0–8, ADG0–12, and ADG0–14), feed conversion ratio (FCR) during 0–4, 0–8, 0–12, 0–14 weeks of age (FCR0–4, FCR0–8, FCR0–12, and FCR0–14), and survival rate during 0–4, 0–8, 0–12, 0–14 weeks of age (SUR0–4, SUR0–8, SUR0–12, and SUR0–14). From each crossbred chicken, artificial insemination was used for breeding chicken cocks and hens at a ratio of 1:5 (chicken cocks:chicken hens). A total of nine mating pairs were arranged, and the number of animals in each mating pair is shown in Table 1. This study was reviewed and approved for institutional animal care based on the Ethics for Animal Experimentation of the National Research Council of Thailand (No. IACUC-KKU-20/65).

### 2.2. Animal Management

The chickens were raised under the open environment system. Each breed was managed separately as follows. During hatching to four weeks of age, all chicks were reared on a cement husk-covered floor with brooders. Then, they were moved to a growing pen with a litter floor and raised in a house open to natural light. The size of the growing pen was 2 × 2 m. The density was approximately eight chicks per square meter, which is able to raise 32 chickens per pen. The incandescent lighting was programed with two stages: the first stage was from hatching to 4 weeks with 24 h light/0 h dark; the second stage was from 4 to 14 weeks (slaughtering weight) with 23 h light/1 h dark.

The hatched chicks were leg banded until 4 weeks of age followed by wing banding to keep their purebred and crossbred records. All chickens received Newcastle vaccination and/or any antibiotics according to the vaccination program for chickens. All chicks were provided water and fed ad libitum using standard commercial broiler feed. The feed was divided into two formulas according to the age of the chickens: from hatching to 4 weeks of age, 21% crude protein (CP) and 3000 kcal metabolizable energy (ME/kg), followed by a diet of 19% crude protein (CP) and 2900 kcal metabolizable energy (ME/kg).

### 2.3. Characteristics of Carcass and Meat

At 14 weeks of age with slaughtering weight, eight chickens (four male and four female chickens) of each purebred and crossbred were randomly selected and dissected for carcass analysis; thus, 72 samples were used. First, they were fasted for 12 h and then weighed individually. The chickens were slaughtered and dressed following the Thai processing style [34]. Individual carcass data were taken for each chicken as follows: (1) live weight (g); (2) eviscerated weight was measured after removal of the head and the intestine; (3) carcass yields were calculated in both as actual weight (g) and percentage (%) by weight of the carcass of the living chicken before slaughter [35].

After determination of the carcass, meat samples of the chicken carcass consisting of breast meat, thigh meat, chicken drumsticks, and chicken wings were cut and weighed. The color measurements of meat and skin were taken from the breast and thigh parts using a Chroma Meter (model CR-400, Minolta Camera Co. Ltd., Osaka, Japan). The lightness (*L**), redness (*a**), and yellowness (*b**) values measurements were identified using the Commission Internationale de l’Eclairage system (CIE) [36]. Averages of *L**, *a**, and *b** values were calculated from three different locations on the surface of the samples.

### 2.4. Statistical Analysis

The growth performance data (body weight, average daily gain, feed conversion ratio, and survival rate), combining ability (GCA, SCA, RCA), and heterosis values, among purebred (HB × HB, CB × CB, TN × TN) and crossbred (HB × TN, TN × HB, CB × HB, HB × CB, TN × CB, CB × TN) chickens were analyzed by multi-factor ANOVA (sex, chicken hatch set and breed) using a general linear model for unbalanced data (GLM procedure) by the SAS package to investigate the significant difference. If significant differences were detected, multiple pairwise comparisons were conducted using *Scheffe’* (*p* < 0.05).

The data of carcass percentages (%carcass, %breast meat, %thigh, %drumstick, and %wing) and color of the meat and skin (breast and thigh) were checked for normality by the Shapiro–Wilk test. The Levene’s test assessed the homogeneity of variance across treatments (purebred and crossbred chickens). If a significant deviation from a normal distribution and/or homogeneity of variance were observed, the nonparametric Kruskal–Wallis ANOVA rank test was applied to determine the differences between the treatment groups. The treatment effects were considered to be significant at *p* < 0.05 using the Dwass–Steel–Critchlow–Fligner test. All data were expressed as mean values with pooled standard errors.

### 2.5. Combining Ability and Heterosis Analysis

The diallel crossing system is an animal mating system in which all animal breeds must have the opportunity to mate with other animal breeds and all mating pairs (in this study, three chicken breeds were able to produce a total of nine crossbreeds). The genetics of each mating pair of chickens were determined through a genetic matching test based on the following values: general combining ability (GCA), a value indicating the cumulative influence of the additive gene effect, which is determined when using both males and females of the same breed; specific combining ability (SCA), a value that indicates the specificity of genetic matching; and reciprocal combining ability (RCA), which represents the specificity of the genetic pairing when alternating between male and female breeds; heredity is the opposite of the SCA value.

For the combining ability analysis, GCA, SCA, and RCA followed the fixed model of Griffing [32]. The heterosis percentage was analyzed using the mean data over mid-parents and better parents. The equation can be written as follows:(1)$$GCA=\frac{1}{2p}\left(X_{i.}+X_{.i}\right)-\frac{1}{p^{2}}X..$$
(2)$$SCA=\frac{1}{2}\left(X_{ij}+X_{ji}\right)-\frac{1}{2p}\left(X_{i.}+X_{.i}+X_{j.}+X_{.j}\right)+\frac{1}{p^{2}}X..$$
(3)$$RCA=\frac{1}{2}\left(X_{ij}-X_{ji}\right)$$
where *p* is the number of chicken breeds; $X_{i.}$ is the sum of the mean of trait characteristics between chicken cock breed *i* and other chicken hen breeds; $X_{.i}$ is the sum of the mean of trait characteristics between other chicken cock breeds and chicken hen breed I; $X_{j.}$ is the sum of the mean of trait characteristics between chicken cock breed *j* and other chicken hen breeds, $X_{.j}$ is the sum of the mean of the trait characteristics between other chicken cock breeds and chicken hen breed *j*; $X_{ij}$ is the mean of the trait characteristics between the chicken cock breed and chicken hen breed *ij*; $X_{ji}$ is the mean of the trait characteristics when switching the parent breeds *ji*; and X.. is the sum of the mean of the trait characteristics from all chicken breeds (grand total).

Heterosis was calculated according to Fairfull [37] by SAS of the following formula:(4)$$\%H=\frac{HV}{{\bar{P}}_{S+D}}\times 100$$
where %H is the % of heterosis; HV is the heterosis value; and ${\bar{P}}_{S+D}$ is the average phenotype of chicken cocks and chicken hens.

## 3. Results

### 3.1. Body Weight and Growth Performance

Figure 1 shows the BW, ADG, FCR, and SUR in three different purebreds. There were no statistically significant differences in hatching weight (BW0) among breeds (*p* > 0.05). However, CB × CB presented the highest body weights + SE from 4 (344.08 ± 7.42 g) to 14 (1580.75 ± 30.42 g) weeks of age (*p* < 0.05). The means of ADG and FCR were greater and lower, respectively, in CB × CB than in HB × HB and TN × TN in every week of age (*p* < 0.05). HB × HB was the lightest (*p* < 0.05) and its FCR was the highest (*p* < 0.05). The CB × CB survival rate was lower than that in HB × HB and TN × TN (*p* < 0.05), and the difference was clear when compared with the HB × HB survival rate. Figure 2 presents the BW, ADG, FCR, and SUR in crossbreds. BW0 did not differ among the groups (*p* > 0.05). The highest body weights from 4 to 14 weeks of age, ADG, and survival rates were found in TN × CB (green bar), and CB × TN (blue bar), while those of HB × TN (black bar) and CB × HB (gray bar) were the lowest (*p* < 0.05). The FCRs of TN × CB (green bar) and HB × TN (black bar) were the lowest and highest, respectively (*p* < 0.05).

### 3.2. Combining Abilities and Heterosis Percentage

Three parameters of combining the abilities regarding the general combining ability (GCA), specific combining ability (SCA), and reciprocal effects (RCA) on the BW and ADG are presented in Table 2. The GCA effects were positive and considerably high for TN and CB (*p* < 0.05), ranging from 0.89 to 70.75 g for BW0 to BW14 and 0.02 to 0.71 g/day for ADG 0–4 to ADG 0–14 in TN and ranging from −0.07 to 87.83 g for BW0 to BW14 and 1.23 to 0.90 g/day for ADG 0–4 to ADG 0–14 in CB, but negative for HB, ranging from −0.82 to −158.58 g for BW0 to BW14 and −1.25 to −1.61 g/day for ADG 0–4 to ADG 0–14 (*p* < 0.05). The SCA effect in CB × HB was negative, ranging from −0.03 to −19.75 g, and was positive for CB × TN and HB × TN, ranging from 0.66 to 40.49 g (*p* < 0.05). The RCA effects in TN × HB and HB × CB were negative, ranging from −0.64 to −61.35 g, and positive for TN × CB, ranging from 1.34 to 51.09 g (*p* < 0.05).

The percentage of heterosis for BW and ADG is shown in Figure 3. For the body weight trait (Figure 3a), the positive values at BW4, BW8, and BW12 in the crossbreeds were TN × CB (green line), CB × TN (blue line), TN × HB (orange line), and HB × CB (yellow line). However, the value was negative at all weeks of age in CB × HB (gray line), at BW12 in HB ×TN (black line), and BW14 in HB × CB (yellow line) (*p* < 0.05). For the ADG traits, Figure 3b shows that all crossbreeds decreased with increasing age. TN × CB (green line), CB × TN (blue line), and TN × HB (orange line) were positive values at all ages. However, CB × HB (gray line) was negative at all weeks of age (*p* < 0.05).

### 3.3. Carcass and Meat Characteristics

The carcass characteristics are shown in Table 3. CB × CB (87.00%) produced a higher carcass percentage than TN × TN (85.05%) and HB × HB (82.91%) (*p* < 0.05). For crossbred chickens, the highest carcass percentage was found in TN × CB (89.65%), followed by CB × TN (88.55%), and the lowest carcass percentage was found in TN × HB (71.91%) (*p* < 0.05). The purebred breast, thigh, drum, and wing percentages were higher in CB × CB and TN × TN than in HB × HB (*p* < 0.05). In crossbreeds, TN × CB had a carcass percentage higher than the other crossbreeds, especially in the breast (19.32%), thigh (12.33%), and drumstick (12.21%), followed by CB × TN. The lowest carcass percentage in crossbred chickens was found in TN × HB (*p* < 0.05).

### 3.4. Meat Characteristics

The meat and skin color of the breast and thigh parts are shown in Table 4. In the purebred, the highest breast and thigh meat lightness (*L**) values were obtained in TN × TN (*p* < 0.05). The skin lightness values also followed those of the meat values. In the crossbreeds, HB × CB had the lowest lightness values because of the apparent meat intensity. In addition, melanin pigment was deposited in black-bone chicken meat, and CB × HB had the lowest lightness of meat color. For yellowness (*b**) values in the color of meat and skin of the breast and thigh area, TN had higher yellowness (*b**) compared to other breeds (*p* < 0.05). The redness of meat (*a**) showed that TN was positive for both the meat and the thigh. The results showed that TN had more redness in the meat and skin. However, the other crossbred was negative (*p* < 0.05). The compared sex was found to be negative. The meat and skin color of the breast and thigh of the male chicken had higher brightness (*L**) than the female, while the meat color of the female chicken thigh had a higher yellowness (*b**) than the males (*p* < 0.05).

## 4. Discussion

This research proposed developing and studying the chicken mating system to improve the crossbreeding between black-bone chickens and Thai native chickens for better growth performance and maintaining the color of black-bone chickens to gain desirable antioxidant properties (melanin and carnosine).

The differences observed in the body weights of each purebred can be described by different genetics, which resulted in a difference in their body size and growth performance (Figure 1). The highest GCA was found in CB × CB; meanwhile, GCA was significantly negative in HB × HB of all ages (Table 2). Buranawit et al. [22] reported that the Chinese black-bone chicken’s purebred had a high genetic growth rate. When compared to KU-Phuparn (Thai black-bone chicken) [38], the CB × CB chickens in this research showed a higher body weight at 12 weeks of age. Meanwhile, Thai native chickens have developed a genetic improvement in the optimal market size at 1200 g [13,17]. The Hmong black-bone chicken is a native chicken with a small body size classified as wildfowl or mountain fowl. Vietnam black chickens of mature size have an average body weight in range from 450 to 500 g [39]. HB originated in the north-central region of Thailand. Our researchers have improved their genetic growth performance since 2002. However, HB has a small size with a slow growth rate, and its ability to grow is limited; the matured weight was reported to be range approximately from 1250 to 1550 g [40,41,42,43]. The negative values in HB indicate that the growth genetics were not affected by their growth. In other words, weight gain might have resulted from management rather than genetics.

We found that crossbreeding CB and TN had high growth performance and a low feed conversion ratio (Figure 2), which were related to positive values of their SCA and RCA, respectively (Table 2). The SCA presented a nonadditive genetic effect. In other words, food and management, at least similar to our conditions, mainly impacted the growth performance. Similarly, Siwendu et al. [44] reported that the SCA crossbred of three chickens in South Africa had the highest and most positive effect on SCA related to body weight. However, the RCA value of TN × CB was higher than the SCA value of CB × TN. This fact might be due to the maternal effect of CB being low. Therefore, crossing between the TN should be used as a sire line, while CB should be used as a dam line for the best growth performance. Meanwhile, the crossbreeding of HB to either CB or TN had a low growth performance, which could be explained by the HB’s small size and genetics with a low GCA. Together with the consideration of SCA and RCA, these values were the lowest at all ages, suggesting that HB is not suitable for crossbreeding under a commercial system.

The heterosis percentages on BW and ADG of crossbreeding (Figure 3a,b) demonstrated that both CB × TN and TN × CB were better than their parents, resulting in reaching the market size faster. Based on the results shown in Table 2, the crossbreeding can be selected at 8 weeks of age, which is the optimum age of the preselection genetics for faster growth in crossbreeds to be developed into a crossbred line by the Inter Se mating system. This strategy could reduce the generation interval and farm management and would have the potential for market competition.

Interestingly, the heterosis effect for a crossbred bodyweight between CB sires and HB dams was negative in all weeks of age (−3.00% to −14.08%; Figure 3). The negative values might result from dominant genes, overdominance, or heterozygosity [45,46]. This finding was similar to Siwendu et al. [44], who reported on Ross sires and Naked Neck dams, suggesting that crossbreeding of these two breeds should be avoided.

When considering the carcass characteristics (Table 4), we focused on *L**, which showed the melatonin content index. Zhang et al. [47] reported that skin color is important in black-bone chicken because the color of the meat and skin of the black chicken was used to determine the market price. The black color of the meat caused by the accumulation of melanoproteins and the pigment melanin in mammals and poultry is controlled by the genetics [48]. In addition, melanin binds to oxygen molecules (reactive oxygen species; ROS), producing protective and antioxidant properties [49]. The results showed that crosses between TN × CB and CB × TN were insignificant (*p* > 0.05) but were lighter than purebred CB. However, these crossbred black skin colors could be further improved by using gene markers as a selection tool [19]. Improving the black skin color in high growth performance crossbreeds could help poultry producers meet the consumer demand for meat and health products produced from this chicken population. 

## 5. Conclusions

Several black-boned chicken breeds are thought to have medicinal powers in Asian countries, resulting in high costs; however, the growth performance of purebreds is low. Therefore, crossbreeding is enhanced by increasing the growth rate and maintaining a similar meat quality to that of the black-boned chicken. The diallel crossing system is essential to compare the specific combining ability (SCA) with general combining ability (GCA). Furthermore, the reciprocal combining ability (RCA) is significant in determining the lines of both the sire and dam, resulting in an elite population of hybrids. The present study focused on the crossing between native chicken species and we suggest that crossing between the TN sires and CB dams had the highest potential for growth performance and carcass characteristics. The development of these hybrid chickens is possible for further production and distribution on an industrial scale. 

## Figures and Tables

**Figure 1 animals-12-01602-f001:**
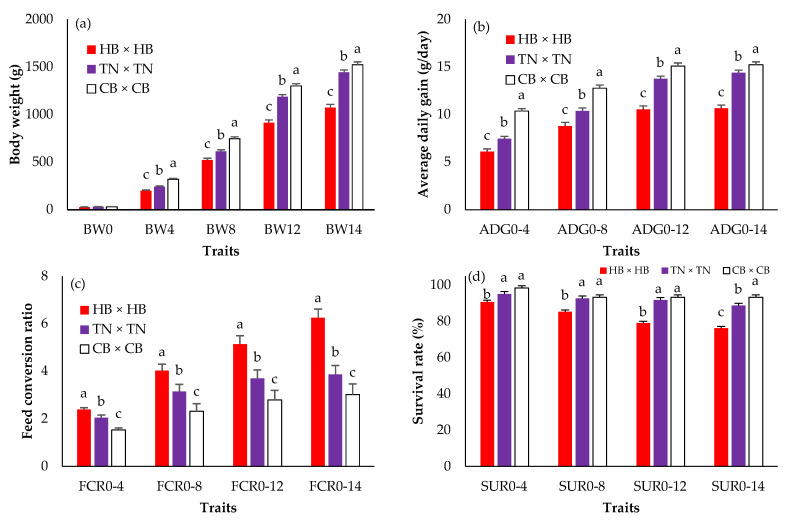
The least squares means (standard error) of the body weight (**a**); average daily gain (**b**); feed conversion ratio (**c**); survival rate (**d**) in purebred black-bone and Thai native chickens; a, b, c: Means for the trait with different letters differ significantly at *p* < 0.05; HB × HB—purebred Hmong black-bone chicken (red bar); TN × TN—purebred Thai native chicken (purple bar); CB × CB—purebred Chinese black-bone chicken (white bar); BW0—birth weight (g); BW4, BW4, BW8, BW12, BW14—birth weight and body weight at 4, 8, 12, 14 (slaughtering weight) weeks of age (g); ADG0–4, ADG0–8, ADG0–12, ADG0–14—average daily gain during 0–4, 0–8, 0–12, 0–14 weeks of age (g/day); FCR0–4, FCR0–8, FCR0–12, FCR0–14—feed conversion ratio during 0–4, 0–8, 0–12, 0–14 weeks of age; SUR0–4, SUR0–8, SUR0–12, SUR0–14—survival rate during 0–4, 0–8, 0–12, 0–14 weeks of age (%).

**Figure 2 animals-12-01602-f002:**
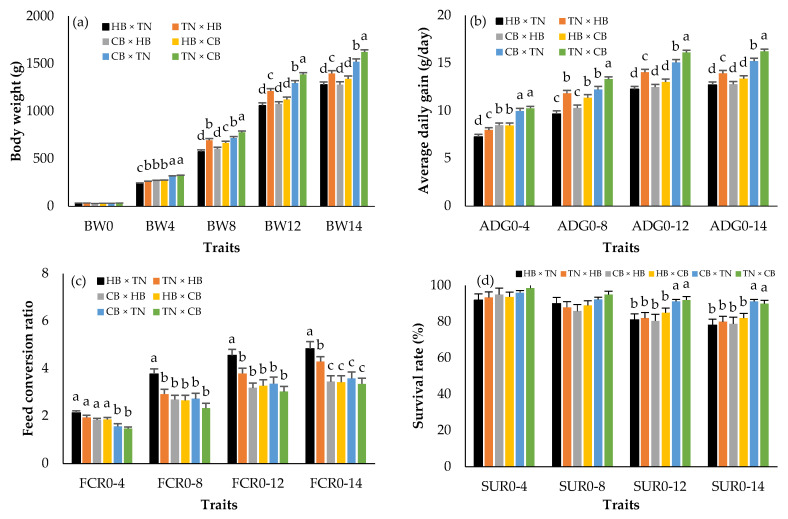
Least squares means (standard error) of body weight (**a**); average daily gain (**b**); feed conversion ratio (**c**); survival rate (**d**) in purebred black-bone and Thai native chickens; a, b, c, d: Means for the trait with different letters differ significantly at *p* < 0.05; HB × TN—crossbred chicken between Hmong black-bone chicken (used as chicken cocks) and Thai native chicken (used as chicken hens) (black bar); TN × HB—crossbred chicken between Thai native chicken (used as chicken cocks) and Hmong black-bone chicken (used as chicken hens) (orange bar); CB × HB—crossbred chicken between Chinese black-bone chicken (used as chicken cocks) and Hmong black-bone chicken (used as chicken hens) (gray bar); HB × CB—crossbred chicken between Hmong black-bone chicken (used as chicken cocks) and Chinese black-bone chicken (used as chicken hens) (yellow bar); CB × TN—crossbred chicken between Chinese black-bone chicken (used as chicken cocks) and Thai native chicken (used as chicken hens) (blue bar); TN × CB—crossbred chicken between Thai native chicken (used as chicken cocks) and Chinese black-bone chicken (used as chicken hens) (green bar); BW0—birth weight (g); BW4, BW4, BW8, BW12, BW14—birth weight and body weight at 4, 8, 12, 14 (slaughtering weight) weeks of age (g); ADG0–4, ADG0–8, ADG0–12, ADG0–14—average daily gain during 0–4, 0–8, 0–12, 0–14 weeks of age (g/day); FCR0–4, FCR0–8, FCR0–12, FCR0–14—feed conversion ratio during 0–4, 0–8, 0–12, 0–14 weeks of age; SUR0–4, SUR0–8, SUR0–12, SUR0–14—survival rate during 0–4, 0–8, 0–12, 0–14 weeks of age (%).

**Figure 3 animals-12-01602-f003:**
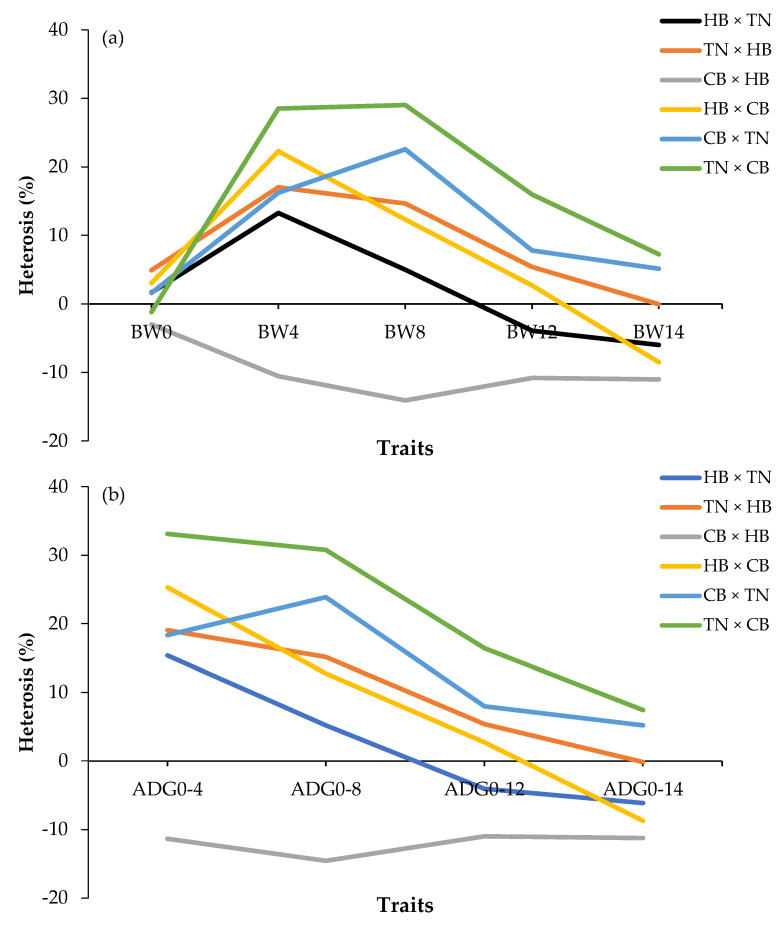
Percentage of heterosis of body weight (**a**); average daily gain (**b**); HB × TN—crossbred chicken between Hmong black-bone chicken (used as chicken cocks) and Thai native chicken (used as chicken hens) (black bar); TN × HB—crossbred chicken between Thai native chicken (used as chicken cocks) and Hmong black-bone chicken (used as chicken hens) (orange bar); CB × HB—crossbred chicken between Chinese black-bone chicken (used as chicken cocks) and Hmong black-bone chicken (used as chicken hens) (gray bar); HB × CB—crossbred chicken between Hmong black-bone chicken (used as chicken cocks) and Chinese black-bone chicken (used as chicken hens) (yellow bar); CB × TN—crossbred chicken between Chinese black-bone chicken (used as chicken cocks) and Thai native chicken (used as chicken hens) (blue bar); TN × CB—crossbred chicken between Thai native chicken (used as chicken cocks) and Chinese black-bone chicken (used as chicken hens) (green bar); BW0—birth weight (g); BW4, BW4, BW8, BW12, BW14—birth weight and body weight at 4, 8, 12, 14 (slaughtering weight) weeks of age (g); ADG0–4, ADG0–8, ADG0–12, ADG0–14—average daily gain during 0–4, 0–8, 0–12, 0–14 weeks of age (g/day).

**Table 1 animals-12-01602-t001:** The diallel crossing system and number of black-bone chickens, Thai native chickens, and their crossbreeds.

Chicken Breed/Sex	Female		
Male	HB	CB	TN
**HB**	GCA_HB×HB_ (n = 55)	SCA_HB×CB_ (n = 86)	SCA_HB×TN_ (n = 79)
**CB**	RCA_CB×HB_ (n = 63)	GCA_CB×CB_ (n = 72)	SCA_CB×TN_ (n = 79)
**TN**	RCA_TN×HB_ (n = 54)	RCA_TN×CB_ (n = 103)	GCA_TN×TN_ (n = 108)

HB—Hmong black-bone chicken; CB—Chinese black-bone chicken; TN—Thai native chicken (Pradu Hang dum); GCA_HB×HB_—general combining ability between purebred Hmong black-bone chicken; GCA_CB×CB_—general combining ability between purebred Chinese black-bone chicken; GCA_TN×TN_—general combining ability between purebred Thai native chicken (Pradu Hang dum); SCA_HB×CB_—specific combining ability between crossbred Hmong black-bone chicken (used as chicken cocks) and Chinese black-bone chicken (used as chicken hens); SCA_HB×TN_—specific combining ability between crossbred Hmong black-bone chicken (used as chicken cocks) and Thai native chicken (Pradu Hang dum) (used as chicken hens); SCA_CB×TN_—specific combining ability between crossbred Chinese black-bone chicken (used as chicken cocks) and Thai native chicken (Pradu Hang dum) (used as chicken hens); RCA_CB×HB_—reciprocal combining ability between crossbred Chinese black-bone chicken (used as chicken cocks) and Hmong black-bone chicken (used as chicken hens); RCA_TN×HB_—reciprocal combining ability between crossbred Thai native chicken (Pradu Hang dum) (used as chicken cocks) and Hmong black-bone chicken (used as chicken hens); RCA_TN×CB_—reciprocal combining ability between crossbred Thai native chicken (Pradu Hang dum) (used as chicken cocks) and Chinese black-bone chicken (used as chicken hens).

**Table 2 animals-12-01602-t002:** The combining ability for body weight and average daily gain of black-bone chickens, Thai native chickens, and their crossbreeds.

Items/ Specification	BW0	BW4	BW8	BW12	BW14	ADG 0–4	ADG 0–8	ADG 0–12	ADG 0–14
	(g)	(g)	(g)	(g)	(g)	(g/day)	(g/day)	(g/day)	(g/day)
**GCA**									
HB × HB	−0.82 ^de^	−30.37 ^d^	−70.95 ^f^	−131.89 ^e^	−158.58 ^f^	−1.25 ^e^	−1.25 ^e^	−1.56 ^e^	−1.61 ^g^
TN × TN	0.89 ^ab^	9.65 ^b^	9.78 ^c^	50.34 ^b^	70.75 ^a^	0.02 ^c^	0.16 ^b^	0.59 ^b^	0.71 ^b^
CB × CB	−0.07 ^c^	40.01 ^a^	61.17 ^a^	81.55 ^a^	87.83 ^a^	1.23 ^a^	1.09 ^a^	0.97 ^a^	0.90 ^a^
**SCA**									
HB × TN	0.69 ^b^	0.45 ^c^	17.49 ^b^	20.45 ^c^	12.26 ^c^	0.19 ^b^	0.30 ^b^	0.23 ^c^	0.12 ^d^
CB × HB	−0.03 ^c^	3.74 ^bc^	−7.07 ^d^	−19.75 ^d^	−12.15 ^d^	−0.14 ^cd^	−0.12 ^c^	−0.23 ^d^	−0.12 ^e^
CB × TN	0.66 ^b^	5.33 ^bc^	14.09 ^b^	40.49 ^b^	33.15 ^b^	0.41 ^ab^	0.26 ^b^	0.49 ^b^	0.35 ^c^
**RCA**									
TN × HB	−1.06 ^d^	−2.14 ^c^	−21.25 ^e^	−41.94 ^e^	−25.81 ^de^	−0.04 ^c^	−0.40 ^d^	−0.51 ^d^	−0.27 ^e^
HB × CB	−0.64 ^d^	−61.35 ^e^	−49.27 ^ef^	−27.84 ^d^	−45.59 ^e^	−0.38 ^d^	−0.87 ^de^	−0.32 ^d^	−0.46 ^ef^
TN × CB	1.34 ^a^	11.44 ^b^	15.56 ^b^	45.05 ^b^	51.09 ^ab^	0.36 ^b^	0.25 ^b^	0.53 ^b^	0.52 ^b^

GCA—general combining ability; SCA—specific combining ability; RCA—reciprocal combining ability; CB—Chinese black-bone chicken; HB—Hmong black-bone chicken; TN—Thai native chicken (Pradu Hang dum); ^a,b,c,d,e,f,g^ Means for the body weight and average daily gain traits in the same column with different letters differ significantly at *p* < 0.05.

**Table 3 animals-12-01602-t003:** The carcass weight and carcass percentage of black-bone chickens, Thai native chickens, and their crossbreds.

Items/ Specification	Body Weight (kg)	Carcass Weight (kg)	Breast (kg)	Thigh (kg)	Drumstick (kg)	Wing (kg)	% Carcass	% Breast	% Thigh	% Drumstick	% Wing
**GCA**											
CB × CB	1.42 ^bc^	1.24 ^c^	0.28 ^b^	0.16 ^cd^	0.17 ^abc^	0.14 ^bc^	87.00 ^a^	19.32 ^ab^	11.33 ^ab^	12.12 ^a^	9.73 ^b^
HB × HB	1.08 ^e^	0.89 ^e^	0.18 ^f^	0.12 ^f^	0.12 ^e^	0.11 ^d^	82.91 ^ab^	16.51 ^d^	10.68 ^b^	11.19 ^a^	10.31 ^ab^
TN × TN	1.53 ^ab^	1.30 ^ab^	0.28 ^ab^	0.17 ^ab^	0.18 ^ab^	0.16 ^a^	85.05 ^ab^	18.37 ^ab^	11.33 ^ab^	11.65 ^a^	10.28 ^ab^
**SCA**											
CB × HB	1.20 ^d^	1.01 ^d^	0.23 ^de^	0.15 ^e^	0.14 ^de^	0.12 ^c^	84.24 ^ab^	19.00 ^ab^	12.13 ^a^	11.44 ^a^	10.20 ^ab^
CB × TN	1.56 ^a^	1.38 ^a^	0.30 ^a^	0.19 ^a^	0.19 ^a^	0.16 ^a^	88.55 ^a^	19.07 ^ab^	12.24 ^a^	12.17 ^a^	10.05 ^ab^
HB × TN	1.30 ^d^	1.03 ^d^	0.24 ^cd^	0.15 ^d^	0.15 ^bcd^	0.14 ^bc^	83.28 ^ab^	18.86 ^ab^	11.93 ^ab^	12.02 ^a^	10.42 ^ab^
**RCA**											
HB × CB	1.29 ^d^	1.07 ^d^	0.22 ^e^	0.16 ^d^	0.15 ^cd^	0.13 ^bc^	83.28 ^ab^	17.23 ^cd^	12.07 ^b^	11.55 ^a^	10.42 ^ab^
TN × CB	1.48 ^abc^	1.32 ^ab^	0.29 ^b^	0.18 ^bc^	0.18 ^bc^	0.15 ^bc^	89.65 ^a^	19.32 ^ab^	12.33 ^a^	12.21 ^a^	10.07 ^b^
TN × HB	1.41 ^c^	1.01 ^d^	0.24 ^c^	0.17 ^c^	0.13 ^de^	0.14 ^ab^	71.94 ^c^	17.07 ^cd^	11.92 ^b^	9.41 ^b^	9.95 ^b^

GCA—general combining ability; SCA—specific combining ability; RCA—reciprocal combining ability; CB—Chinese black-bone chicken; HB—Hmong black-bone chicken; TN—Thai native chicken (Pradu Hang dum); ^a,b,c,d,e,f^ Means for the carcass traits in the same column with different letters differ significantly at *p* < 0.05.

**Table 4 animals-12-01602-t004:** The meat and skin color of the breast and thigh parts in black-bone chickens, Thai native chickens, and their crossbreds.

Items/ Specification	Meat Color						Skin Color					
	*L**		*a**		*b**		*L**		*a**		*b**	
	Breast	Thigh	Breast	Thigh	Breast	Thigh	Breast	Thigh	Breast	Thigh	Breast	Thigh
**GCA**												
CB × CB	43.74 ^c^	29.58 ^c^	−2.36 ^bc^	−2.21 ^b^	9.09 ^d^	3.96 ^bcd^	40.36 ^c^	37.35 ^c^	−1.81 ^cde^	−2.53 ^b^	5.60 ^e^	3.21 ^d^
HB × HB	45.53 ^bc^	29.85 ^c^	−1.64 ^bc^	−2.23 ^b^	10.77 ^cd^	2.63 ^d^	39.12 ^c^	34.56 ^cd^	−2.17 ^de^	−2.26 ^b^	5.95 ^e^	4.21 ^d^
TN × TN	51.80 ^a^	54.44 ^a^	0.79 ^a^	2.19 ^a^	15.59 ^a^	12.69 ^a^	55.91 ^a^	60.76 ^a^	0.22 ^a^	2.51 ^a^	18.42 ^a^	18.28 ^a^
**SCA**												
CB × HB	49.88 ^ab^	33.18 ^b^	−1.34 ^bc^	−1.18 ^b^	12.52 ^bc^	5.53 ^b^	45.71 ^b^	40.51 ^b^	−1.63 ^cd^	−2.30 ^b^	8.79 ^d^	4.96 ^cd^
CB × TN	53.32 ^a^	36.15 ^b^	−3.23 ^c^	−1.29 ^b^	13.00 ^b^	4.78 ^bc^	45.04 ^b^	41.38 ^b^	−0.77 ^b^	−2.20 ^b^	11.06 ^c^	6.27 ^c^
HB × TN	52.24 ^a^	36.42 ^b^	−2.25 ^bc^	−2.00 ^b^	12.67 ^bc^	4.37 ^bc^	44.39 ^b^	40.48 ^b^	−1.45 ^bcd^	−2.58 ^b^	8.43 ^d^	4.49 ^d^
**RCA**												
HB × CB	38.23 ^d^	27.91 ^c^	−2.67 ^bc^	−2.07 ^b^	5.98 ^e^	3.45 ^cd^	35.46 ^d^	33.97 ^d^	−1.67 ^cd^	−2.62 ^b^	5.77 ^e^	3.98 ^d^
TN × CB	50.49 ^a^	36.74 ^b^	−1.18 ^b^	−1.67 ^b^	13.08 ^b^	4.77 ^bc^	47.25 ^b^	41.08 ^b^	−1.14 ^bc^	−1.37 ^b^	12.34 ^c^	8.88 ^b^
TN × HB	53.14 ^a^	34.95 ^b^	−2.72 ^bc^	−1.97 ^b^	13.68 ^ab^	5.05 ^bc^	46.33 ^b^	41.90 ^b^	−2.43 ^e^	−2.86 ^b^	14.88 ^b^	4.41 ^d^
**SEX**												
Male	50.22 ^a^	36.35 ^a^	−1.54 ^a^	−1.48 ^a^	12.20 ^a^	4.69 ^b^	45.45 ^a^	42.96 ^a^	−1.56 ^a^	−1.90 ^a^	9.96 ^a^	6.29 ^a^
Female	46.89 ^b^	34.59 ^b^	−2.14 ^a^	−1.27 ^a^	11.43 ^b^	5.81 ^a^	43.35 ^b^	39.71 ^b^	−1.29 ^a^	−1.70 ^a^	10.31 ^a^	6.76 ^a^

GCA—general combining ability; SCA—specific combining ability; RCA—reciprocal combining ability; CB—Chinese black-bone chicken; HB—Hmong black-bone chicken; TN—Thai native chicken (Pradu Hang dum); ^a,b,c,d,e^ Means for the meat and skin color of breast and thigh traits in the same column with different letters differ significantly at *p* < 0.05; *L**—Lightness; *a**—redness; *b**—yellowness.

## Data Availability

The data presented in this study are available upon request from the Network Center for Animal Breeding and Omics Research, Faculty of Agriculture, Khon Kaen University, Thailand.

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
