# Peer review of "Combining Abilities, Heterosis, Growth Performance, and Carcass Characteristics in a Diallel Cross from Black-Bone Chickens and Thai Native Chickens"

_animals, 2022, doi:10.3390/ani12131602_

Round 1

Reviewer 1 Report

The paper examines the combining abilities, heterosis, growth performance, and carcass characteristics in a diallel cross from black-bone chickens and Thai native chickens. The topic is very relevant since there is a growing demand for poultry production worldwide. The paper presents novel and useful findings. The introduction provides evidence-based background for the research. The methods have been properly described, results are well presented and data interpretation is appropriate. The findings are thoroughly discussed, and conclusions are justified by the results. I did not find any objective errors. 

There is a minor suggestion that is incorporated directly into the text, several references have to be added and some minor explanations to be provided.

All the best and stay safe

Author Response

Dear Reviewer,

Response to Reviewer 1 Comments

The paper examines the combining abilities, heterosis, growth performance, and carcass characteristics in a diallel cross from black-bone chickens and Thai native chickens. The topic is very relevant since there is a growing demand for poultry production worldwide. The paper presents novel and useful findings. The introduction provides evidence-based background for the research. The methods have been properly described, results are well presented and data interpretation is appropriate. The findings are thoroughly discussed, and conclusions are justified by the results. I did not find any objective errors. 

There is a minor suggestion that is incorporated directly into the text, several references have to be added and some minor explanations to be provided.

All the best and stay safe

Response: We are grateful for the critical reading and your efforts to improve the quality of the manuscript.  We hope that the manuscript in its revised form will please you. We edited the revised manuscript as your suggestions such as fixing the errors as your suggested, added more references that related to this research, and explained further in your questions. Please see the revised manuscript.

Best Regards,

Wuttigrai

Reviewer 2 Report

Manuscript describes important problem increase of growth rate in case of Hmong black-bone, Chinese black-bone and Thai native chickens using heterosis effect, what allow to use such lines after crossing as slow-growing broiler chickens. Additionally meat of those chickens can be an important source of carnosine (Tian et al. 2007; Eur. Food. Res. Technol.).

The text is legible, interesting and almost clear to understand, but I have a few comments/suggestions:

Line 2 and line 3: I don’t know if it is manuscript’s fault but better is not to separate words in title (hard space could be useful: shift+ctrl+space)

Line 23: lack of number of observations in case of BW and ADG (perhaps due to mortality of chickens, maybe better is to use range from 699 at the beginning of experiment to … in week 14)

Line 26: p<0,05

Line 71: ‘They have desirable antioxidant properties’ this is not clear. Maybe better precise: melanin in skin, carnosine in meat, and both also decrease cholesterol level in blood

Line 77: appreciated by consumers instead of outstanding (subjective opinion)

Line 131: Experimental design is not clearly explained. By 4 weeks chickens were kept together, so then they were allocated to pens 2x2 meters with surface of 1m2  (for 8 chickens as one replication?). In this case in each treatment should have different number of replications, but in case of one pen  not equal number of chickens should be observed?

Line 145: After random selection data had normal distribution for each treatment/experimental group (Shapiro-Wilk test) and homogenous variances (Leven test)?. In case of  699 chickens in 9 treatments  there is no problem because power of a test is high

Line 212: no significant differences in BW0 of chickens between 3 purebreds were observed. I propose without  letters (a)

Line 224: the same like in line 212 (without letters a)

Line 225: the same like in line 212 with SUR0-4 and SUR0-8

Line 264: Items/Specification instead of Parameters Chickens

Line 295: Table 3 could be arrange the same like table 2. In case of weight better will be to use one decimal place (grams as a unit) or present data in kilograms, ie. 1.42 ; 1.08 ; 1.53 … and so on. Kilograms give the same number of decimal places like percentage values in right side of Table 3.

Line 315: the same like line 264 (Table 2)

Line 334: ‘…an average body weight of 450-500 g’. Maybe better will be: in range from 450 to 500 g.

Line 338: the same like in line 334. 1,250-1,550 suggest range of BW, approximately rather one value

Line 348: ‘Therefore we suggest that the CB…’ this sentence sounds like a conclusion, maybe better to change it with sentence from line 379 ‘Crossing between the CB…”

Line 373: insignificant instead of not different

Line 398: Please to check references according to instruction for authors in Animals Journal  (mainly abbreviations of Journals with dots: line 400, 409 and so on )

Author Response

Dear Reviewer,

Response to Reviewer 2 Comments

Manuscript describes important problem increase of growth rate in case of Hmong black-bone, Chinese black-bone and Thai native chickens using heterosis effect, what allow to use such lines after crossing as slow-growing broiler chickens. Additionally meat of those chickens can be an important source of carnosine (Tian et al. 2007; Eur. Food. Res. Technol.).

The text is legible, interesting and almost clear to understand, but I have a few comments/suggestions:

Response: We are grateful for the critical reading and your efforts to improve the quality of the manuscript.  We hope that the manuscript in its revised form will please you. We edited the revised manuscript as your suggestions as follows:

Point 1: Line 2 and line 3: I don’t know if it is manuscript’s fault but better is not to separate words in title (hard space could be useful: shift+ctrl+space)

Response 1: We have edited the title name without no separate words. See lines 2-3.

Point 2: Line 23: lack of number of observations in case of BW and ADG (perhaps due to mortality of chickens, maybe better is to use range from 699 at the beginning of experiment to … in week 14)

Response 2: We edited the sentence “Total data were 699 recorded at the beginning of the experiment to 595 recorded in weeks 14 of age. Body weight (BW), average daily gain (ADG), feed conversion ratio (FCR), and survival rate (SUR) were recorded.” in our manuscript already. See lines 23-25.

Point 3: Line 26: p<0,05

Response 3: We added p<0.05 in revised manuscript. See line 27.

Point 4: Line 71: ‘They have desirable antioxidant properties’ this is not clear. Maybe better precise: melanin in skin, carnosine in meat, and both also decrease cholesterol level in blood

Response 4: We have revised for clearer as follows “They have desirable bioactive compounds including melanin and carnosine which are antioxidant properties, and both also decrease cholesterol levels in the blood [18-21].” See lines 68-70.

Point 5: Line 77: appreciated by consumers instead of outstanding (subjective opinion)

Response 5: We agree with your suggestion. It has been edited as “Thai native chicken has appreciated by consumers for the flavor and texture of tender meat” according to your suggestion. See line 74.

Point 6: Line 131: Experimental design is not clearly explained. By 4 weeks chickens were kept together, so then they were allocated to pens 2x2 meters with surface of 1m2  (for 8 chickens as one replication?). In this case in each treatment should have different number of replications, but in case of one pen not equal number of chickens should be observed?

Response 6:  We have rewritten the animal management on their housing as follows: “The chickens were raised under the open environment system. Each breed was managed separately as follow. During hatching to 4 weeks of age, all chicks were reared on a cement husk-covered floor with brooders. Then, they were moved to a growing pen with a litter floor and raised in a house open to natural light. The sized of the growing pen was 2×2-meter. The density was approximately 8 chicks per square meter which is able to raise 32 chickens per pen. The incandescent lighting was programed consisted of two stages: the first stage was from hatching to 4 weeks with 24 h light/0 h dark; the second stage was from 4 to 14 weeks (slaughtering weight) with 23 h light/1 h dark.” See lines 133-140.

Point 7: Line 145: After random selection data had normal distribution for each treatment/experimental group (Shapiro-Wilk test) and homogenous variances (Leven test)?. In case of  699 chickens in 9 treatments  there is no problem because power of a test is high

Response 7: We carry out the statistical tests as you suggested and have clearly explained in the statistical analysis section. See lines 174-180.

Point 8: Line 212: no significant differences in BW0 of chickens between 3 purebreds were observed. I propose without  letters (a)

Response 8: We already deleted the letters (a) in BW0 from Figure 1. See Figure 1.

Point 9: Line 224: the same like in line 212 (without letters a)

Response 9: We already deleted the letters (a) in BW0 from Figure 2. See Figure 2.

Point 10: Line 225: the same like in line 212 with SUR0-4 and SUR0-8

Response 10: We already deleted the letters (a) in SUR0-4 and SUR0-8 from Figure 2. See Figure 2.

Point 11: Line 264: Items/Specification instead of Parameters Chickens

Response 11: We already changed the column name from “Parameters Chickens” to “Items/Specification” as your suggestion, See line 284.

Point 12: Line 295: Table 3 could be arrange the same like table 2. In case of weight better will be to use one decimal place (grams as a unit) or present data in kilograms, ie. 1.42 ; 1.08 ; 1.53 … and so on. Kilograms give the same number of decimal places like percentage values in right side of Table 3.

Response 12: We already changed the column name from “Parameters Chickens” to “Items/Specification”. In addition, we have changed the display of units of numbers from grams to kilograms and used two decimal places to correspond to the numbers on the right. See Table 3.

Point 13: Line 315: the same like line 264 (Table 2)

Response 13: We already changed the column name from “Parameters Chickens” to “Items/Specification” as your suggestion, See line 316.

Point 14: Line 334: ‘…an average body weight of 450-500 g’. Maybe better will be: in range from 450 to 500 g.

Response 14: We already changed the words from “…an average body weight of 450-500 g” to “an average body weight in range from 450 to 500 g”. See lines 361-362.

Point 15: Line 338: the same like in line 334. 1,250-1,550 suggest range of BW, approximately rather one value

Response 15: We already changed the words from “…an average body weight of 1,250-1,550 g” to “an average body weight in range from 450 to 500 g”. See lines 365.

Point 16: Line 348: ‘Therefore we suggest that the CB…’ this sentence sounds like a conclusion, maybe better to change it with sentence from line 379 ‘Crossing between the CB…”

Response 16: Thank you very much for your point of view. we agree with your comment, and we already changed it please see lines 375-376.

Point 17: Line 373: insignificant instead of not different

Response 17: We already changed the word as your suggestion. See line 400.

Point 18: Line 398: Please to check references according to instruction for authors in Animals Journal  (mainly abbreviations of Journals with dots: line 400, 409 and so on )

Response 18: We already checked and edited references according to instruction for authors in Animals Journal. See Reference section.

Best Regards,

Wuttigrai

Reviewer 3 Report

Dear editors and authors,

I have completed the review of the article titled “Combining Abilities, Heterosis, Growth Performance, and Carcass Characteristics in a Diallel Cross from Black-bone Chickens and Thai Native Chickens”. I must say that it is really fluent and understandable. Thanks to the authors for that. I think it would be appropriate to publish it after a little editing in the introduction.

Best regards,

Reviewer

Introduction

Line 43-67: Increasing demand for poultry meat, production, consumption and many statistics are highlighted. However, I think that this information is given too much due to the subject of the study. This section can be shortened a little.

Line 68-81: In this section, the characteristic features of native chickens are emphasized and appropriate.

Line 82-88: It is understood that very few points have been mentioned about Combining Abilities, Heterosis, Growth Performance, and Carcass Characteristics, which is the main subject of the study. Therefore, instead of more general information (such as production consumption statistics), the focus should be on the main topic. More studies on this subject should be included.

Line 91-94: The purpose of ths stusy is clearly stated.

Materials and Methods

Available.

Results

Clear and available.

Discussion

Clear and available.

Conclusions

Available.

Author Response

Dear Reviewer,

Response to Reviewer 3 Comments

Dear editors and authors,

I have completed the review of the article titled “Combining Abilities, Heterosis, Growth Performance, and Carcass Characteristics in a Diallel Cross from Black-bone Chickens and Thai Native Chickens”. I must say that it is really fluent and understandable. Thanks to the authors for that. I think it would be appropriate to publish it after a little editing in the introduction.

Best regards,

Reviewer

Response: We sincerely thank the reviewer for their dedication and time spent reviewing our research and for realizing its usefulness.

Introduction

Point 1: Line 43-67: Increasing demand for poultry meat, production, consumption and many statistics are highlighted. However, I think that this information is given too much due to the subject of the study. This section can be shortened a little.

Response 1: We have rewritten the sentences to be concise and shorter. See lines 45-64.

Point 2: Line 68-81: In this section, the characteristic features of native chickens are emphasized and appropriate.

Response 2: Thank you very much for your reviewed.

Point 3: Line 82-88: It is understood that very few points have been mentioned about Combining Abilities, Heterosis, Growth Performance, and Carcass Characteristics, which is the main subject of the study. Therefore, instead of more general information (such as production consumption statistics), the focus should be on the main topic. More studies on this subject should be included.

Response 3: We added the sentences that related combining ability and heterosis including references. See lines 85-92.

Point 4: Line 91-94: The purpose of this study is clearly stated.

Response 4: Thank you very much for your reviewed.

Point 5: Materials and Methods

Available.

Response 5: Thank you very much for your reviewed.

Point 6: Results

Clear and available.

Response 6: Thank you very much for your reviewed.

Point 7: Discussion

Clear and available.

Response 7: Thank you very much for your reviewed.

Point 8: Conclusions

Available.

Response 8: Thank you very much for your reviewed.

Best Regards,

Wuttigrai

Reviewer 4 Report

The reviewer thinks that this manuscript is not suitable for publishing in Animals, because Audiences concerned about Thai native chicken, its commercial use are limited, others around the world, except for Thai, can't use that technology, and the results are meaningless for the outsiders at all. It's the primary reason to judge rejection. This decision may be overturned if relevant information for scientists in other countries can be provided and added.

*Introduction

The reviewer carefully read Introduction. It is difficult for the reviewer to comprehend the objective of this research in the end. Could the authors make the objective simpler? Do the authors want to develop and provide new F1 commercial chickens for Thai market?

*Materials and Methods

The reviewer believes that the authors made some mistakes in choosing a statistical method. For example, scientists must not use Duncan’ new multiple range test as post-hoc ANOVA test. The authors should use the combination of ANOVA and Scheffe’s multiple comparison procedure.

2.2. Characteristics of carcass and meat: although the authors described that at 14 weeks of age with slaughtering weight, eight chickens (4 male and 4 female chickens) of each purebred and crossbred were “randomly selected” and dissected for carcass analysis, there are several contradictions between body weight shown in Figure 1 and Figure 2, and body weight shown in Table 3. In the case that samples per test group is small, e.g., carcass and meat traits, scientists must choose nonparametric procedures for group comparison. Therefore, It is essential and necessary to revise materials and methods, and to re-calculate and show the results of carcass and meat characteristics.

*Conclusions

Based on the data provided, the reviewer is of the opinion that TNxCB has consistently demonstrated superior growth potential. It is not clear why the authors emphasize that CBxTN has the greatest potential for growth traits.

Author Response

Dear Reviewer,

Response to Reviewer 4 Comments

The reviewer thinks that this manuscript is not suitable for publishing in Animals, because Audiences concerned about Thai native chicken, its commercial use are limited, others around the world, except for Thai, can't use that technology, and the results are meaningless for the outsiders at all. It's the primary reason to judge rejection. This decision may be overturned if relevant information for scientists in other countries can be provided and added.

Response: We are grateful for the critical reading and your efforts to improve the quality of the manuscript.  We hope that the manuscript in its revised form will please you. For your reviewed and concluded that this research is only useful for poultry production in Thailand, we would like to explain the results and application of this research on various occasions:

  1. Actually, not only Thailand that has its own native and black-bone chickens, but also many countries around the world have their own native chickens such as China, South-Korea, Africa, and Europe with different names. Therefore, applying knowledge of crossbreeding mating systems could enhance their production. Besides, the heterosis of crossbreeding would be a new opportunity to develop a new genetic chicken line that has more potential for competition in terms of healthy food with growth performance in the poultry industry in the future. It also helps reduce costs and increase profits for farmers more than ever.
  2. Moreover, the main results in the present study demonstrate the importance of animal breeding development for improving the low performance and low productivity in livestock animals until to branding. In other words, the research must take into account the benefits to other groups of consumers as well, such as livestock business operators and health-conscious consumers. Therefore, it is necessary to think beyond publication in order to reach practical use in the livestock business, which will help the research solve livestock problems in the entire supply chain system.

Point 1: *Introduction

The reviewer carefully read Introduction. It is difficult for the reviewer to comprehend the objective of this research in the end. Could the authors make the objective simpler? Do the authors want to develop and provide new F1 commercial chickens for the Thai market?

Response 1: Your understanding is right. The objective of this study was to evaluate the appropriate mating pairs for growth performance traits between black-boned chickens (Chinese black and Hmong) and Thai native chickens (Pradu Hang dum) to develop and provide new F1 commercial chickens for Thai market. We have revised the objective simpler as your suggestion. See lines 94-97.

Point 2: *Materials and Methods

The reviewer believes that the authors made some mistakes in choosing a statistical method. For example, scientists must not use Duncan’ new multiple range test as post-hoc ANOVA test. The authors should use the combination of ANOVA and Scheffe’s multiple comparison procedure.

Response 2: We confirmed that the research has been planned and statistically correctly, which can be described as follows: In this research, we identified chicken pairs (9 pairs in both purebred and crossbred chickens) as the treatment factor (single factor), meanwhile the observation value consists of growth performances (body weight, average daily gain, feed conversion ratio, and survival rate), carcass and meat characteristics. Therefore, the combination of ANOVA (multiple factors) was not used in this statistical analysis. In the case of the treatment mean analysis, both Duncan’s new multiple range test (DMRT) and Scheffe' test are also effective methods for testing the difference in treatment means, especially the DMRT method is very popular in animal science research, for this reason the authors chose the DMRT method in this study.

Point 3: 2.2. Characteristics of carcass and meat: although the authors described that at 14 weeks of age with slaughtering weight, eight chickens (4 male and 4 female chickens) of each purebred and crossbred were “randomly selected” and dissected for carcass analysis, there are several contradictions between body weight shown in Figure 1 and Figure 2, and body weight shown in Table 3. In the case that samples per test group is small, e.g., carcass and meat traits, scientists must choose nonparametric procedures for group comparison. Therefore, It is essential and necessary to revise materials and methods, and to re-calculate and show the results of carcass and meat characteristics.

Response 3: We confirmed that the body weight of the chickens in all purebred and crossbred chickens in Figure 1, Figure 2, and Table 3 are consistency because chickens were from the same sample throughout the experiment. However, it could be possible that the simple random sampling method in the present study resulted in contradictions between body weight shown in Figure 1 and Figure 2, and body weight shown in Table 3.

However, for carcass and meat experiment, we agree with your suggestion and have reanalyzed using nonparametric procedures (the nonparametric Kruskal-Wallis ANOVA rank test) as shown in Table 3. 

Point 4: *Conclusions

Based on the data provided, the reviewer is of the opinion that TNxCB has consistently demonstrated superior growth potential. It is not clear why the authors emphasize that CBxTN has the greatest potential for growth traits.

Response 4: It is truth as your stated that CBxTN has the greatest potential for growth traits. We apologized for our mistake in conclusion, the revised conclusion has been corrected. See lines 37-39 in the abstract and line 406-407 in the conclusion.

Best Regards,

Wuttigrai

Round 2

Reviewer 4 Report

The reviewer maintains the first judgement of rejection, because in the response to the reviewer's comments, it is more clear that this manuscript is an essay for Thai complacency, by Thai, and for the Thai market.

Authors need to be aware that the reviewer is a statistical expert. In particular, the following characterization is totally unacceptable.   

Response 2: We confirmed that the research has been planned and statistically correctly, which can be described as follows: In this research, we identified chicken pairs (9 pairs in both purebred and crossbred chickens) as the treatment factor (single factor), meanwhile the observation value consists of growth performances (body weight, average daily gain, feed conversion ratio, and survival rate), carcass and meat characteristics. Therefore, the combination of ANOVA (multiple factors) was not used in this statistical analysis. In the case of the treatment mean analysis, both Duncan’s new multiple range test (DMRT) and Scheffe' test are also effective methods for testing the difference in treatment means, especially the DMRT method is very popular in animal science research, for this reason the authors chose the DMRT method in this study.

Author Response

Dear Reviewer,

We have revised our manuscript follow by your suggestions please see the information below for details:

Point 1: The reviewer maintains the first judgement of rejection, because in the response to the reviewer's comments, it is more clear that this manuscript is an essay for Thai complacency, by Thai, and for the Thai market.

Response 1: Thank you for pointing out the ambiguity of our manuscript, which could be misunderstood for further application in terms of “this work is an essay only for Thai complacency, by Thai, and for the Thai market.” We sincerely apologize for disagreeing with the reviewer on your main reason for rejection. At least researchers who are interested in black-bone chicken development, could have an idea of crossbreeding between black-bone and native breeds from our study to apply to their work. Actually, we mentioned the importance of utilizing the diallel crossing system in the introduction section (lines 85-92). However, we have now also added the scope of application of our work in the conclusion section (lines 398-404) to demonstrate that breeding methods for the diallel crossing system should be considered for the genetic improvement of the desired crossbred animals.:  

               “Several black bone chicken breeds are thought to have medicinal powers in Asian countries, resulting in high costs; however, the growth performance of purebreds is low. Therefore, crossbreeding with native chicken is one of the solutions to reduce cost with a possibility of retaining meat quality. The diallel crossing system is essential to compare specific combining ability (SCA)with General combining ability (GCA). Besides, the reciprocal combining ability (RCA) is significant in determining sire and dam lines, resulting in elite population hybrids.”

Point 2: Authors need to be aware that the reviewer is a statistical expert. In particular, the following characterization is totally unacceptable.   

Response 2: We confirmed that the research has been planned and statistically correctly, which can be described as follows: In this research, we identified chicken pairs (9 pairs in both purebred and crossbred chickens) as the treatment factor (single factor), meanwhile the observation value consists of growth performances (body weight, average daily gain, feed conversion ratio, and survival rate), carcass and meat characteristics. Therefore, the combination of ANOVA (multiple factors) was not used in this statistical analysis. In the case of the treatment mean analysis, both Duncan’s new multiple range test (DMRT) and Scheffe' test are also effective methods for testing the difference in treatment means, especially the DMRT method is very popular in animal science research, for this reason the authors chose the DMRT method in this study.

Response 2: The other concern is the method used for multiple comparisons; therefore, we reanalyzed data with multi-factor (breed, sex and hatch factor) as you suggested using Scheffe’ instead of DMRT. The results of reanalyzing demonstrated the less liberal of differences among pairs of comparisons. We also revised the tables and results (lines 165-178, Figure 1, Figure 2, Figure 3, Table 2, Table 3, and Table 4).

Sincerely yours,

Wuttigrai Boonkum,

Corresponding’ author
